# RealNodes: Interactive and Explorable 360° VR System with Visual Guidance User Interfaces

## ABSTRACT

Emerging research expands the idea of using 360-degree panoramas of the real-world for "360 VR" experiences beyond video and image viewing. However, most of these are strictly guided, with few opportunities for interaction or exploration. There is a desire for experiences with cohesive virtual environments with choice in navigation, versus scripted experiences with limited interaction. Unlike standard VR with the freedom of synthetic graphics, there are challenges in designing user interfaces (UIs) for 360 VR navigation within the limitations of fixed assets. We designed RealNodes, a novel software system that presents an interactive and explorable 360 VR environment. We also developed four visual guidance UIs for 360 VR navigation. The results of a comparative study of these UIs determined that choice of user interface (UI) had a significant effect on task completion times, showing one of the methods, Arrow, was best. Arrow also exhibited positive but non-significant trends in preference, user engagement, and simulator-sickness. RealNodes and the comparative study contribute preliminary results that inspire future investigation of how to design effective visual guidance metaphors for navigation in applications using novel 360 VR environments.

**Keywords:** Immersive / 360° video; 3D user interaction; Non-fatiguing 3DUIs; Locomotion and navigation; 3DUI metaphors; Computer graphics techniques

**Index Terms:** Human-centered computing—Human computer interaction (HCI)—Interaction paradigms—Virtual Reality; Human-centered computing—Interaction design —Interaction design process and methods—User interface design;

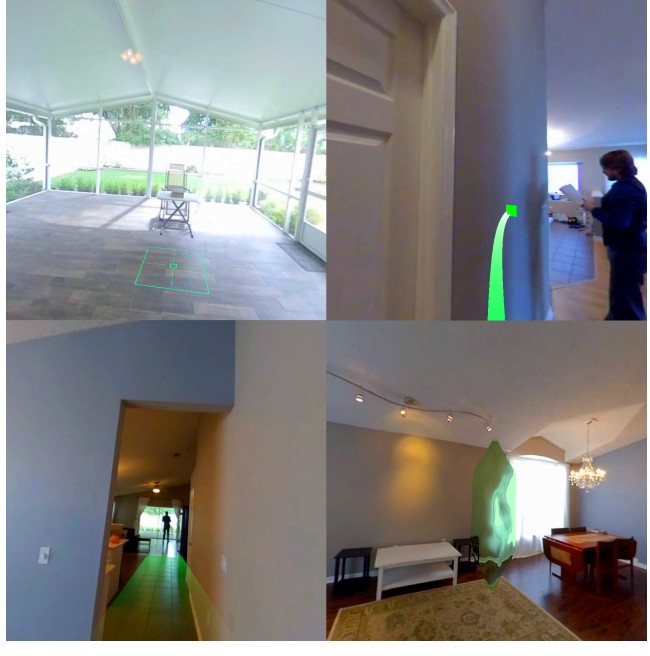

Figure 1: Images from the RealNodes software displaying the four types of visual guidance UI. (Top-left) Target; (Top-right) Arrow; (Bottom-left) Path; (Bottom-right) Ripple.

## 1 INTRODUCTION

Recent virtual reality (VR) research has reignited interest in using 360-degree panoramas of real-world environments for "360 VR" experiences. Using image panoramas for 360-degree VR experience dates to QuickTime VR [4]. In the late 90's and early 2000s, some video games experimented with making worlds with synthetic 360-degree environment maps, connected such that if a user clicked in certain directions, they would go to another location [21] [2]. Only recently with affordable consumer 360-degree cameras have both technology and research advanced to explore developing rich 360 VR environments like these using real-world images.

Experiences in development today are expanding beyond video and image viewing, exploring technological and human factors challenges of enhancing immersion beyond guided experiences. Muhammad et al. [16] explored using Walking-In-Place (WIP) locomotion to control 360-video playback, finding that simulator sickness was reduced compared to passive playback. Lin et al. developed two focus assistance interfaces for watching 360-video [13], finding their methods improved ease of focus overall, but other positive effects depended on video content and viewer goals. MacQuarrie and Steed [14] developed VEs from connected 360-images and three visual transitions, finding that methods with animation gave

a better feeling of motion. Rhee et al. developed MR360, software demonstrating real-time integration of interactive objects in a 360-video live stream, accurately lit with Image Based Lighting (IBL) improving presence compared to conventional 360-videos [20].

One problem needing further investigation is appropriate user interface (UI) metaphors for visual guidance, waypoint finding, and navigation geared to 360 VR's limitations. Synthetic environments have the freedom of granular user position and geometry, while 360 VR is limited by fixed assets for environments. Emerging applications would benefit from this: training systems with navigation-based decision-making; virtual tours with freedom take branching paths; improved fitness applications with multiple routes chosen during runtime instead of in a menu; novel games that incorporate 360 VEs made with real-world capture data for unique experiences.

To tackle this, we developed a novel software system called RealNodes, an engine for scenarios combining 360-degree video and virtual assets to create *Nodes*, or 360 VR location waypoints, connected in a cohesive interactive environment. For this research, we implemented four visual guidance UIs for indicating waypoints in 360 VR: Target, Ripple, Path, and Arrow (pictured in Fig. 1).

A comparative study was performed on the UIs. Participants explored a 360 VR scenario and performed a searching task in four conditions, each with a different UI and hidden object location. We present preliminary results that show one of the methods, Arrow, had a statistically significant difference in scenario competition times. Conditions with Arrow had significantly faster completion times, more than two times faster than the slowest condition, Path. This

seems to indicate Arrow is easier to learn and use in the scenarios. Participant testimonials and other metrics give more explanation for why the Arrow could be preferred for 360 VR. We contribute a design for a software system and a set of visual guidance UIs for the application space of 360 VR. We additionally provide preliminary results of a comparative evaluation which provide feedback and inspiration into designing and refining UIs for 360 VR.

## 2 RELATED WORK

We discuss four major categories in 360 VR: Navigation Techniques, Assisted Focus and Guidance for Wayfinding, Effective Visual Transitions, and Integration of Interactive Elements. We feel these are relevant to 360 VR navigation and interaction, and demonstrates how we build upon and differentiates from prior research.

### 2.1 Navigation Techniques

Navigation in VR is widely researched, providing effective options for users to explore virtual environments (VEs). Machuca et al. [3] devised a UI called Fluid VR to perform navigation and selection task controls with a seamless transition between modes. They claim that traditional VR applications with explicit button or gesture mode switching have the propensity for mode errors, which can frustrate users. To solve this, they developed three constraint-based interaction techniques: manipulation, pre-programmed interaction-based navigation, and flying. They claim high accuracy and ease of use, allowing for fluid mode changes. RealNodes has two modes: navigation and interaction, toggled with an explicit button press. However, we provide clear visual indication of mode change with the Visual Guidance UI preventing false mode input.

Tanaka et al. [22] devised a large-scale experiment to test a novel "Motive Compass" slide pad interface with arrows and velocity control for navigating 360 VR on a mobile touch device, compared to a conventional on-screen button-based method. Their results show that their interface better presents accessible directions compared to conventional methods. Though RealNodes does not use Motive Compass since it is an HMD based system, it has taken inspiration from both the visuals used in their experiment for visual guidance metaphors as well as the ability to control speed of navigation.

Paris et al. [18] developed two user studies comparing methods of exploring large virtual worlds in the constraint of a smaller physical space and their effect on presence and spatial memory in a navigation and memorization task. Their second study compared two methods of reorientation: one with artificial rotational gain, and the other with distractors. Results indicated a significant effect on learning the environment, with scenarios using rotational gain taking less time than the distractor method. RealNodes has discrete, user-centered locations that do not require much real-world locomotion, instead using a WIP navigation method as a compromise to provide the feeling of walking. However, a future version could be developed that ties navigation transition videos to real locomotion for use in a larger physical space, requiring reorientation or redirection.

Muhammad et al. [16] explored using WIP to navigate 360 VR by controlling playback of a video in 360 VR. They found that compared to passive interfaces that merely played back video, simulator sickness was reduced. One limitation was that rotation caused false positives in their step algorithm. We overcame that problem in RealNodes by allowing steps only when rotated towards a navigable path. Another limitation that Muhammad et al. had was that the application was limited to linear playback, compared to RealNodes which allows for choice in direction.

### 2.2 Assisted Focus and Guidance for Wayfinding

In the realm of 360-video, a constant challenge is encouraging focus on intended targets in the video and not letting the viewer feel like they missed something important. This has led to work on UIs that either directly change focus or more gradually guide viewers to points of interest in the video. To do this effectively, we need to look at prior literature for wayfinding in VEs. Freitag, Weyers, and Kuhlen [7] devised an interactive assistance interface that guides the user to unexplored areas on request. Their methods renders colored tube paths on the ground for a set of locations. This method is an automatic system determined by heuristics factoring in what locations were already seen/explored and what places are interesting based on viewpoint quality. Their focus was on the concept of "exploration success", which they define as seeing all important areas in an environment. A user study was performed to determine if the proposed method added a benefit compared to free exploration without the interface. They found an improvement in knowledge of the environment and higher exploration success, and participants experienced the interface as helpful and easy to use. Their solution was designed for arbitrary scenes without the need to modify a scene with landmarks. Meanwhile RealNodes has a similar but stricter restriction by only having real-world video data and no environment geometry. RealNodes is concerned with exploring effective graphical assisted guidance for exploration success like the work of Freitag et al. Our Arrow method was in part inspired by their method of always indicating direction of close by waypoints.

Lin et al. developed focus assistance user interfaces and ran a comparative experiment to determine methods for visually guiding users while they watch a 360-video, in response to the challenge of continuous focus and refocus on intended targets [13]. Their two techniques were "Auto Pilot", which directly changes the view to the target, and "Visual Guidance", which indicates the direction of the target. They found their focus assistance techniques improved ease of focus overall, but any other effects focus assistance has depended on both the video content and the goals for watching the video. This content was developed for standard 360-video streaming, but lessons learned from this inspired our work to improve navigation and interaction tasks in 360 VR with our comparative study on methods targeted at a multi-path navigation and searching task.

### 2.3 Effective Visual Transitions

A challenge in VR is visual presentation of location transitions. 360 VR adds the additional challenge of lacking virtual world geometry. Moghadam and Ragan [15] conducted a comparative study on three visual transitions for movement in VR: Teleportation, Animated interpolation, and Pulsed interpolation (a series of intermediary teleportations). Results show faster transitions are preferred by participants with VR experience, while slower transitions are preferred by those who do not. Based on this work, we made considerations about speed and interpolation for RealNodes system of visual transitions that blends 360-video of the current location with a manually filmed transition to the next waypoint.

Cho et al. [5] developed a novel method to integrate many 360-images into a VE by using simple image interpolation and warping to simulate movement from different locations as a user navigates. Our method differs in having a VE made from videos connected with ground truth filmed transitions.

MacQuarrie and Steed [14] developed VEs by connecting 360-degree images and implemented ray cast point-and-click navigation. A comparative study was done on three transitions: instantaneous teleport, linear movement through 3D reconstruction, and an image based "Möbius" transformation. Results indicate that 3D model and Möbius were better for feeling of motion. Taking inspiration from this work but in lieu of 3D models, RealNodes implements manually filmed transitions to give a similar feeling of motion. While their system only had one kind of metaphor for waypoints (floating spheres), we implemented four types of visual guidance UI.

### 2.4 Integration of Interactive Elements

Authoring 360 images and video with virtual elements is an emerging research challenge. Felinto et al. [6] developed a production

framework for combining 360 images with synthetic elements. They incorporated IBL to give synthetic objects reflections and shadows matching the environment. They use "Support" objects to match the position of a real-world object in the image, texture mapped to blend into the scene and add physics and deformation. Though RealNodes requires manual authoring of scenarios compared to the framework made by Felinto et al., it does so with both video and image data.

Rhee et al. developed a software system called MR360 to demonstrate real-time integration of interactive virtual objects in a 360-video live stream [20]. It uses the input video to apply IBL to objects, including realistic shadows from generated light sources. A user study showed improved presence compared to conventional 360-videos without interaction. RealNodes employs IBL to light any virtual objects like MR360 to match the environment. MR360 was designed to generate VEs with live 360-videos, compared to the RealNodes system which uses prerecorded images and video.

## 3  SOFTWARE DESIGN - REALNODES

RealNodes is a novel software system for creating immersive and interactive 360 VR scenarios, developed to explore the challenge of effective navigation in 360 VR. It presents separate locations as their own 360-video, each logically connected with 360-video transitions facilitating multi-path, bidirectional traversal.

For our work, we utilized the Insta360 One X commodity 360-degree camera (See Fig. 2) [1]. It is a two-lens multi camera system in a small form factor. Each wide-angle lens captures 200 degrees FOV, providing a 10-degree buffer of overlap around each lens to aid in stitching. It has onboard buttons for configuration and recording, but we used the official Android app to allow remote start/stop of recording, which avoids having the cameraperson in the videos. After several test recordings of our scenario environment, we chose to record all our video with the 3840 * 1920 @ 50 FPS ("4K") mode, balancing high resolution with relatively smooth playback in VR.

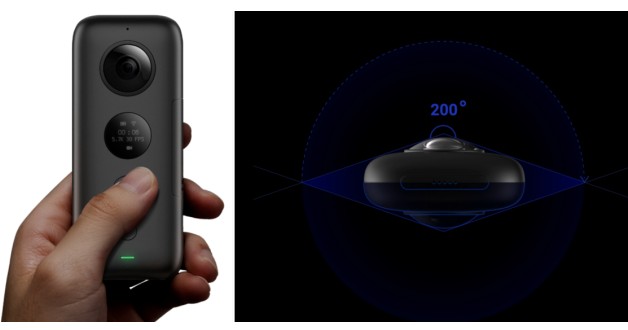

Figure 2: Side and top view of Insta360 One X and viewing angles.

In lieu of a standard tripod, we made our own monopod from the Insta360 One X selfie stick and a ruggedized mini tripod intended for large professional desktop cameras. This minimized the camera appearance in the nadir of the 360-degree video, all while standing as stable as a conventional tripod. Though such monopods for this purpose are sold commercially, this setup worked for us.

RealNodes was made for SteamVR platforms, primarily the HTC Vive. RealNodes minimally requires one controller and uses only two buttons: *Trigger* and *Grip*. Using the *Trigger* simulates a grabbing action. This is used to either grab objects or to manipulate switches in the environment. The *Grip* button is used as a mode toggle to turn *NavigationMode* On/Off. This controls both the display of visual guidance UIs as well as the ability to use WIP.

### 3.1  Software Design Overview

RealNodes was developed with Unity3D to run on Windows 10 desktops. Since most of the scene is defined from 360-videos, there are fewer objects in the scene graph compared to most VEs. Instead, these are dense with scripting logic to define the visuals and control state based on user action. Fig. 3 illustrates the high-level architecture of RealNodes, showing the relationship between major scripts. *MultiVideoManager* is the densest script, defining the logic for video playback and rendering. *PathHotspotV2* is the state machine for each *Waypoint* object. *ViveCursor* controls ray casting logic for the HMD, which indicates which waypoint the user is facing. *SensorDataManager* is the focal point of sensor tracking data and button states. *WalkInPlaceScript* gets raw HMD tracking data and processes it to determine whether WIP is being performed. *ComplexSwitch* tracks the state of an *ActionTool*. *SpawnPageAnimated* handles both the spawning and pickup behavior of a *Page* object. *CsvBuilder* is used to write timestamped metrics to log files for experimental data collection purposes: *SensorLog* which records the tracking and button state data, and *EventLog* which records when a new transition is started and when the scenario ends.

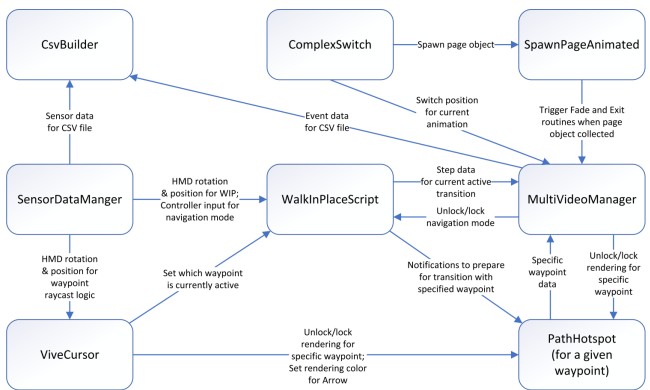

Figure 3: High-level architecture diagram of the different script modules in RealNodes and their inputs/outputs to each other.

### 3.2  Video Manager

A novel feature of RealNodes is a multi-video management system for changing the active video based on event triggers and providing layered video effects. Unity3D Render Textures are used as targets for our videos. A Render Texture can be modified and updated at run time. This allows for receiving dynamic data like video frames while being treated as a standard texture by our *SkyboxPanoramicBlended* shader and merged with graphical effects that are finally sent to the user's view. *MultiVideoManager* holds references to Unity3D Video Player objects, all the video files, and all the Render Textures. Our system allows for smoothly fading between two videos as they are simultaneously playing, such as the current location video and a transition video. It also supports preprocessed videos with a green "alpha" mask color to place an animation on top of the background. This allows for animating partial regions of video based on event triggers, such as opening a closet on command.

Fig. 4 illustrates the high-level render pipeline for the Video Manager. At any given time, there are up to three videos loaded in memory, tied to one of three Video Players: *activePlayer*, *nextPlayer*, and *overlayPlayer*. Every frame of the videos is sent to Render Textures, which acts as input to *SkyboxPanoramicBlended*. The image data combined with instructions from *MultiVideoManager* dictates the final environment background. *MultiVideoManager* can intervene at each level of this process: which video clips are

loaded, which videos are playing or paused, and how much of each video is visible.

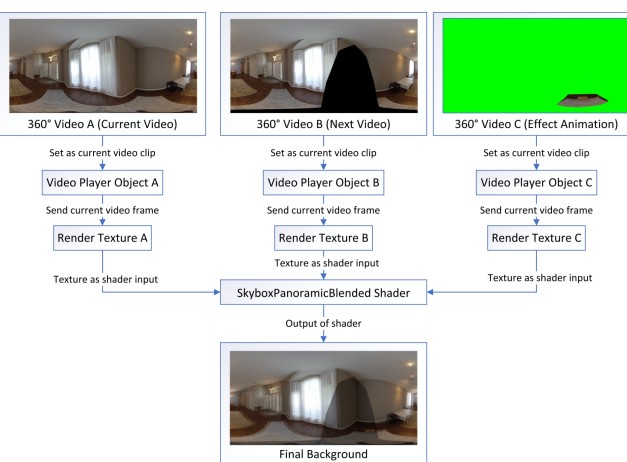

Figure 4: The Video Manager render pipeline, illustrating the high-level process of how the 360 VR environment is made.

*SkyboxPanoramicBlended* is an extension of Unity3D's standard *Skybox − Panoramic* shader. It takes an equirectangular image and renders it as a skybox environment map around the user, using a standard algorithm for converting equirectangular coordinates to spherical coordinates. We make several novel additions. Rather than just one image, it uses three textures: two standard environment maps, and one overlay to the environment map. The custom fragment shader combines this data into the final environment map in addition to a blending coefficient, allowing for fade and overlay effects.

The *activePlayer* plays the primary video for the current situation. Whenever a new video is needed, it is swapped from *nextPlayer* into *activePlayer*. The *nextPlayer* contains the next planned video. Rather than load video when a change is needed, RealNodes dynamically loads the next possible transition video every time the user looks at a new *Waypoint*. This makes it so WIP is initiated with a transition video already in memory. The *overlayPlayer* is loaded with the effect animation for the current *Node*. Unlike the other videos, playback is controlled by the *ActionTool*. When there is no animation for a *Node*, the *overlayPlayer* Render Texture is cleared and set to green, making the system ignore the *overlayPlayer*.

### 3.3   Walking-in-Place Implementation

Our *WalkInPlaceScript* is inspired by the process described by Lee et al. [12]. It tracks sinusoidal motion of the HMD over an input window. It counts steps when a lower peak occurs within a recognition range, correlating with a step. A novel change is that instead of step data being applied as a continuous virtual velocity change, *MultiVideoManager* converts it into playback frames for the current transition video. This allows WIP to realistically match playback, creating the illusion of walking in 360 VR. Visual guidance and WIP can be enabled/disabled by toggling *NavigationMode* by pressing the *Grip* button, reducing false-positive steps.

There are two phases of the WIP: Calibration and Recognition. Calibration consists of two steps: calculating the central axis of the WIP cycle based on the user's default position and applying a recognition range based on that central axis for allowing the WIP to occur. The central axis is computed based on Y-axis height and X-axis rotation (for variance in head pitch) of the HMD. Once Calibration is done, the Recognition loop starts. It collects raw Y-axis position data which is then averaged with a sliding $k$-size window filter. The results are inserted into an $n$-size queue. The

$n$ and $k$ are such that $n > k$, $k \leq 1$, and $n \leq 3$. When there is new data and the filtered queue is full, it checks if the bottom peak of a sinusoidal motion happened. This happens when the middle value is the smallest and the absolute value of the difference between smallest and largest value is more than the recognition range. Once a step is detected, we increase a movement time value to aid in calculating the length of WIP. Each frame, we check if movement time is available. If so, it sends the info to *MultiVideoManager* which plays frames for that amount of time.

### 3.4   Visual Guidance User Interface Methods

We implemented four different visual guidance UIs into RealNodes to indicate where *Waypoint* objects are in the environment and when a user can perform WIP to navigate to those locations. Visual guidance UI display logic is primarily dictated by *PathHotspotV2*, which controls rendering of the UI and signals *MultiVideoManager* to prepare and execute a video transition sequence. Note that for all except the *Arrow* method, the visual guidance is rendered when *NavigationMode* is active, and a user faces the waypoint.

The Target method indicates waypoints with a square shaped, semi-transparent target aligned with the ground plane. The floor Target is inspired by waypoints in conventional VR, often used for teleport navigation [11]. See Fig. 5 for an example, where the next *Node* is in the hallway in front of the door. This allow the Target to indicate an absolute location for where a user can go.

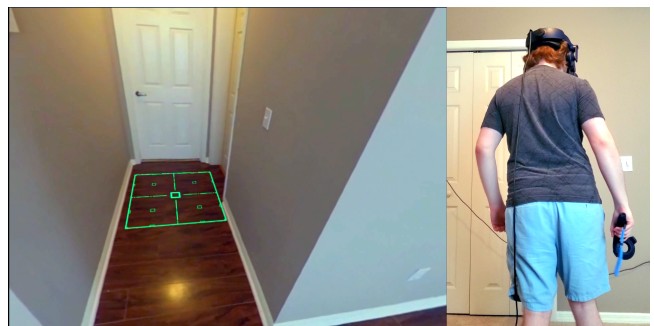

Figure 5: Example of Target UI.

The Ripple method indicates waypoints with a diamond shaped floating marker exhibiting a semi-transparent "ripple" visual effect. We were curious about how a guidance method with a "distortion" effect would affect engagement in a positive way, or if it would negatively impact usability. See Fig. 6 for an example, where the next *Node* is located next to the table. Ripple acts as an absolute waypoint marker in this way. It is animated with a combination of *MaskedRipple* shader and *SimpleTextureScroll* script. *MaskedRipple* is an extension of Unity3D's standard *GlassStainedBumpDistort* shader, taking a normal map texture and distortion coefficient to create a transparent distorted effect. A novel addition we make is a black and white culling mask texture, which acts as a stencil on the render to give it a hexagonal shape. Additionally, *SimpleTextureScroll* manipulates the X and Y texture offsets of the normal map every frame, creating an animation like rippling water.

The Path method indicates the direction of a waypoint with a semi-transparent "lane" aligned with the ground plane and originating from the user. The Path was inspired by the idea of showing a direct line-of-sight path from the user's current location to the waypoint, partly inspired by visualizations from Tanaka et al. [22] of ground-plane based indicators of where a user can navigate. See Fig. 7 for an example, where it is pointing in the direction of the next waypoint, heading into the kitchen. The Path renders as a quad that is rotated

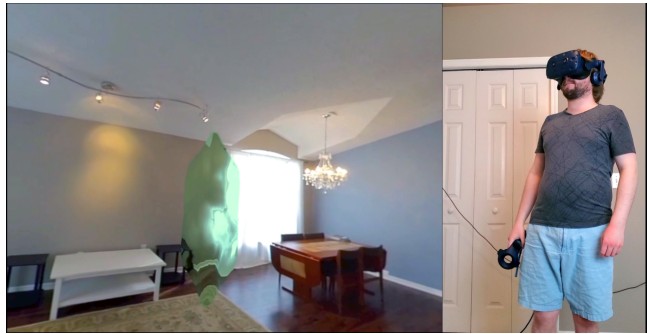

Figure 6: Example of Ripple UI.

based on a the HMD rotation, using *SimpleFixRotation* to keep all but the y axis fixed such that the path is parallel to the ground plane.

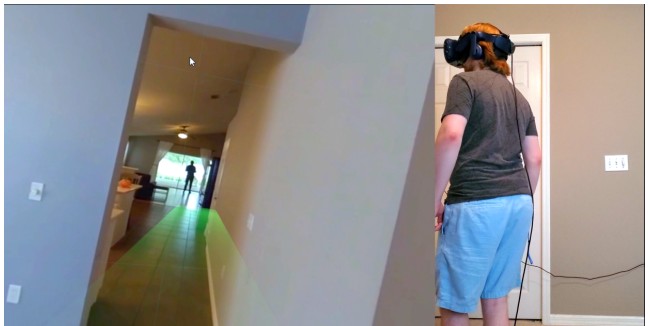

Figure 7: Example of Path UI.

The Arrow method indicates the directions of a waypoint with an arrow formed with a Bezier curve and an arrowhead. Both components actively and smoothly curve towards and point at the nearest waypoint based on shortest rotation distance from user to waypoint. See Fig. 8 for an example of the Arrow UI. When *NavigationMode* is active, the Arrow is always being rendered. When the user is facing a waypoint, the Arrow changes from blue to green. The implementation of Arrow is inspired by guidance methods explored in the work of Lin et al. for indicating points of interest in standard 360-video [13]. The Arrow rendering logic is composed of a Unity3D Line Renderer that draws the arrow curved line, and an arrowhead object. These are controlled by three scripts: *BezierCurve*, *PointArrow*, and *ArrowColor*. *BezierCurve* controls the position of the control points of the Line Renderer that draws the Bezier curve of the Arrow. To match the behavior of *BezierCurve*, *PointArrow* calculates the closest waypoint target by rotation angle between user and waypoint, then smoothly rotates and translates the arrowhead towards that waypoint. Finally, *ArrowColor* changes the color of the Arrow UI based on whether the user is facing a waypoint.

### 3.5 Interaction Objects

As previously mentioned, RealNodes uses IBL from the 360-video data to light the iterative virtual objects. Some interactable objects can manipulate the 360 VR environment. To trigger the animations, we developed an *ActionTool*, a linear drive switch that appears as a golden sphere on a green rail. The golden sphere can be grabbed by hovering the virtual controller over it and holding the *Trigger*. While the *Trigger* is held, the sphere can freely slide along the rail. These *ActionTool* objects are placed in locations where animations

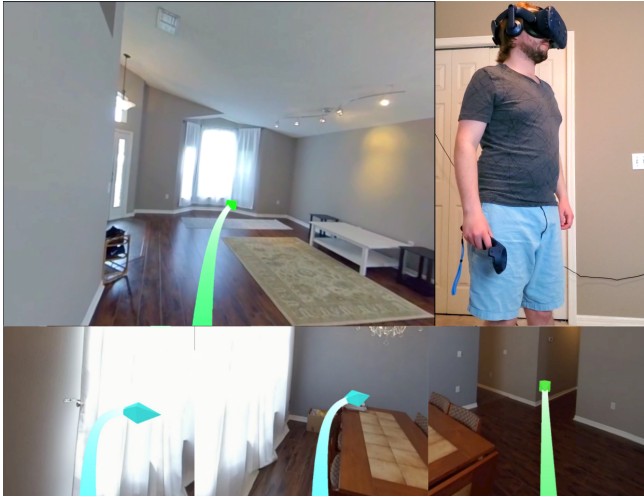

Figure 8: (Top) Example of Arrow UI indicating the direction of the next waypoint is by the window. (Bottom) Arrow curving to nearest waypoint, changing color when nearly facing it.

can be triggered. These animations can reveal hidden objects, such as the collectible *Page* object. This effect is used in the House Scenario developed for our experiment. It includes animations such as opening a drawer/book and lifting a portion of a rug. See Fig. 9 for examples.

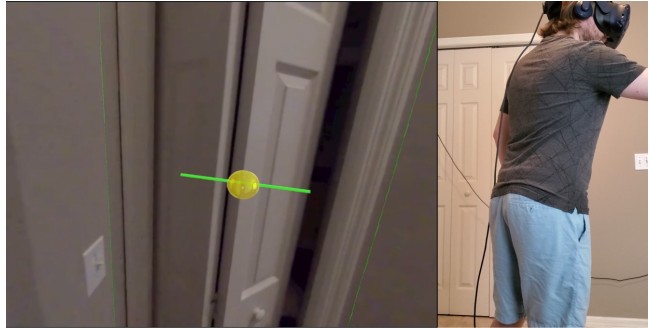

Figure 9: Footage of closet being opened with Action Tool, showing how the masked video is applied to the background and manipulated.

*ComplexSwitch* is attached to each *ActionTool* and controls the state logic of the switch. It tracks the position based on a percentage between the start and end positions. It signals *MultVideoManager* every time position changes, which indicates that animation should play until a specific stopping frame. This results in smooth playback of the animation tied to how much or little the switch is moved. As a usability feature, is dynamically positioned close to the user to make it easier to reach. Once the animation is done, it spawns the *Page* object associated with that animation.

Finding and collecting a *Page* object is the completion goal of the scenario. It appears in the environment when an *ActionTool* is fully switched and a background animation is done. Once it appears, it floats towards the user until it is close enough to be in arm's reach. It can then be collected by hovering the virtual controller over it and pressing the *Trigger* (see Fig. 10).

Each *Page* has a *SpawnPageAnimated* script, which floats the page towards the user and executes the scenario exit conditions

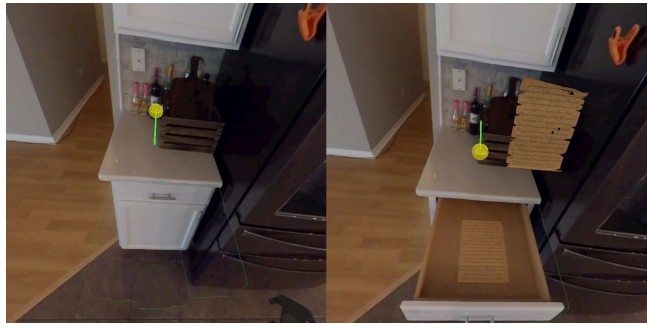

Figure 10: Left is original state of drawer and Action Tool; right is after activating Action Tool, with drawer open and Page object revealed.

when grabbed. On each update frame of the animation, it linearly interpolates a smooth trajectory until it is a specified distance from the user. When the page is collected, the page object is destroyed, signaling *MultiVideoManager* to do a fade transition from the current location video to a black screen, then close the application.

### 3.6 Scenario Development

Developing a 360 VR scenario is an interdisciplinary process requiring skills in film and video editing. RealNodes is no exception to this. It requires planning like pre-production on a video or film, including location scouting to determine an area that fits for the need. Many steps require both manual authoring and software modifications. In the future, authoring tools are planned. Nonetheless, a scenario can be made in short time, helped by reusable scripts and objects. For our study, we developed the House Scenario (see Fig. 11).

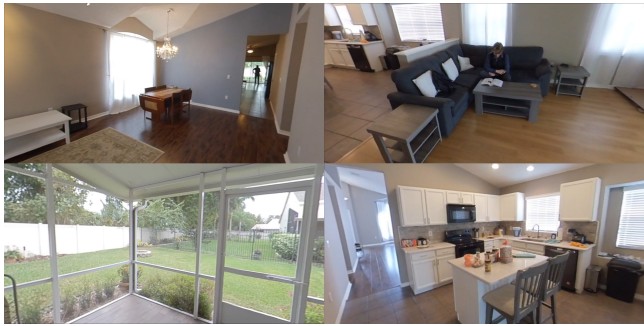

Figure 11: Various scenes from RealNodes House Scenario.

A rough map of the scenario needed to be created to define constraints (see Fig. 12), including locations, routes, and animation locations. From this, we created a list of shots in three categories: Locations, Transitions, and Actions. Locations define a *Node* and are longer videos that needed to roughly loop in the final shot. If an actor was in the shot their motions were planned for the loop (walking back and forth from, reading a book, etc.). These recordings were done remotely without a cameraperson in the shot, adding realism. Transitions require the camera be walked directly from one location to another. To mitigate the presence of the cameraperson, two transitions were filmed for each direction. Actions are short videos of an object moving, used in our animation system. These were filmed after the corresponding Location video to match camera position. We used a common practical effect trick of tying/ hooking a string on an object and pulling on it from a distance, to avoid

having a cameraperson in the final shot. Not counting test filming, 55 shots totaling 13:38 minutes of video are included in the scenario.

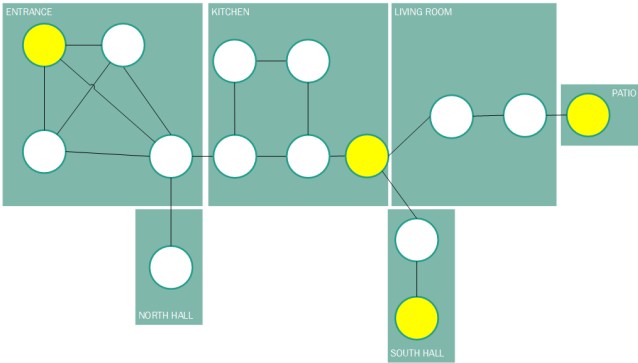

Figure 12: Simple map showing scenario. Circles in the Node graph indicate locations that could be navigated to. Yellow circles indicate locations of hiding spots.

Proprietary format videos produced by Insta360 One X needed to be exported to MP4 using the free companion software Insta360 Studio. All videos were further edited in DaVinci Resolve editing software. Transition and Action video types required masking a region of the clip using a custom mask. For Transitions we mask the cameraperson with a black region, determining that most users would be looking in the direction they are walking during a transition. For Actions, we applied a mask to moving object so everything outside was green, hiding the person pulling the string. See Fig. 13 for an example. After editing is finished, we import the videos into the RealNodes Unity3D project. We chose to transcode all videos to 75% of the resolution (2880 * 1440) to address performance on minimum spec hardware for running our study.

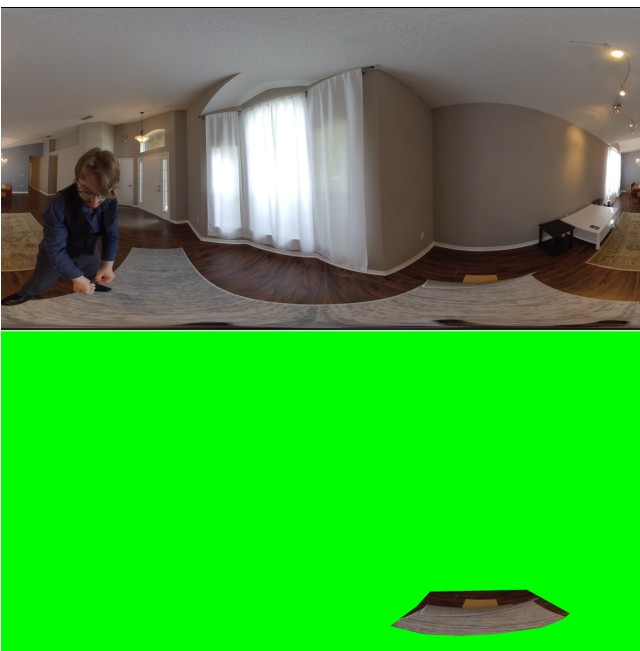

Figure 13: (Top) Someone pulling a string to lift the carpet up. (Bottom) Same as above, except all but the lifted carpet is masked in green.

The amount of work to create a new scenario scene in Unity3D is reduced since most scene objects are the same between scenarios, with the exception of *Nodes*. Each *Node* must be manually populated with *Waypoint*, *ActionTool*, and *Page* objects. These are reusable object instances that already contains support structures and scripts for working with the WIP and visual guidance UI systems (see Fig. 14). An additional benefit to development is instead of requiring *Nodes* to be accurately placed in the scene, our system swaps *Nodes* and their *Waypoints* data relative to user centered position, meaning only *Waypoints* need to be accurately placed.

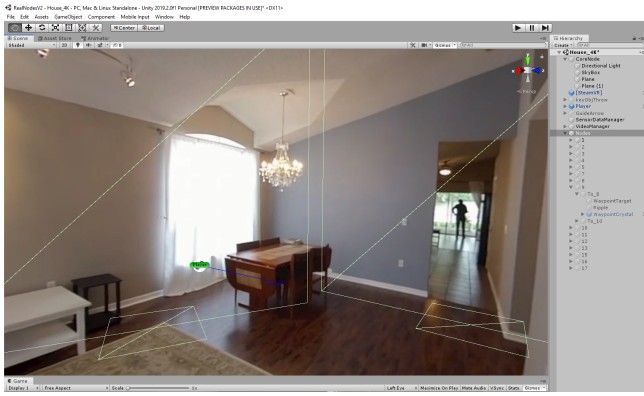

Figure 14: Wireframes of the Waypoint objects in Unity3D editor. Note the large quad perpendicular to the ground plane. This is the target surface for raycasting to detect which waypoint is being faced.

The final steps is minor modification of variable references with scenario specific data. *MultiVideoManager* has public definitions in the editor for its video arrays. *CommonEnums* has enumerations for all videos are stored. These must be modified to match. Then two state machine functions in *MultiVideoManger* for Transitions and Action videos must be modified. In total, the scenario development steps took about 3 days of labor, with preproduciton and filming taking one day, and the rest (editing, waypoint placement, minor code changes) taking the remaining time.

## 4 COMPARATIVE USER STUDY

An experimental study was carried out using the House Scenario in RealNodes to determine the effect of visual guidance UIs on user engagement, simulator sickness, and completion times in 360 VR. The study was performed with 24 people from a volunteer pool of a local university's students, faculty, and staff. The participants were only included if they were physically able to use the HTC Vive HMD and a controller, and jog in place for a duration of a few seconds. The gender breakdown was 18 males and 6 females. Out of 24 participants, 17 wore glasses/contacts. The education breakdown was 12 with high school degree, 8 with undergraduate degree, and 4 with graduate degree. The RealNodes software was run on a Windows 10 desktop computer in a lab setting, with an Intel Core i7-4790K CPU, NVIDIA GeForce GTX 1080 GPU, and 16 Gigabytes (GB) of RAM.

### 4.1 Procedure, Design, and Analysis

After the participant filled out a demographic survey, they were instructed on the controls of RealNodes. They were taught the concept of WIP and how to navigate the environment with it. They were told how to do tasks with hand gestures like grabbing and pulling, and how to switch *NavigationMode* on/off. For each condition, they navigated the environment to find a hidden *Page*. They were told the only way to uncover it was to manipulate an *ActionTool*. Once

the *Page* was grabbed the scenario would end. There were four scenarios, each with a unique visual guidance UI and unique hiding spot. After each condition, they filled out two questionnaires: Simulator Sickness Questionnaire to determine the absence or presence of simulator sickness, nausea, and ocular motor issues (SSQ) [10], and User Engagement Scale Short Form (UES-SF) to determine perception of user engagement and subscores for Aesthetic Appeal (AE), Perceived Usability (PU), Focused Attention (FA), and Reward (RW) [17]. After the last scenario, they filled out a UI preference questionnaire, where 1 = most preferred, and 4 = least preferred.

UI order was randomized across subjects in a counterbalanced manner such that we tested all 24 possible permutations. A Shapiro-Wilk test was performed on Preference, SSQ, UES SF, and completion times to determine normality. The data was non-normalized, so we analyzed using non-parametric tests. A Friedman test was performed on SSQ, UES SF, and Completion times data to determine statistically significant differences between conditions. When an asymptotically significant result was found, we performed post-hoc analysis using a Wilcoxon Signed Ranks Test on all pairs to determine where significance was, followed by a Holm's sequential Bonferroni adjustment to correct for type I errors. For all our statistical measures, we used $\alpha = 0.05$.

### 4.2 Results

We can report that our visual guidance UIs have a statistically significant effect on Completion Times, indicating that Arrow was significantly better at completing our scenario quickly. We additionally found that Arrow had the highest average scores in all but one subscore in preference, simulator sickness, and user engagement, though they were not statistically significant.

#### 4.2.1 Completion Times

We found a statistically significant difference in completion times after performing a Friedman test ($\chi^2(3, 24) = 12.75$, $p < 0.005$). After performing post-hoc analysis using Wilcoxon Signed Rank Test on all possible pairs, a significance was found with two of the pairwise tests: Arrow compared to the Path($Z = -3.686$, $p < 0.001$) and Ripple compared to Path ($Z = -2.029$, $p < 0.05$). Table 1 shows the test statistics for this test.

Table 1: Results of Wilcoxon Signed Rank Test on all pairs.

| Test | Result |
|---|---|
| Target-Ripple | $Z = -0.743$, $p = 0.458$ |
| Target-Path | $Z = -1.914$, $p = 0.056$ |
| Target-Arrow | $Z = -1.4$, $p = 0.162$ |
| Ripple-Path | $Z = -2.029$, $p < 0.05$ |
| Ripple-Arrow | $Z = -0.771$, $p = 0.44$ |
| Path-Arrow | $Z = -3.686$, $p < 0.001$ |

After performing a Holm's sequential Bonferroni adjustment against actual significance threshold, only the Arrow to Path pairwise was found to be significant and the Ripple to Path pairwise test was not, indicating that there was only a signficant difference found between Arrow and Path ($Z = -3.686$, $p < 0.001$).

The mean and standard deviations for completion times (seen in Fig. 15) for each condition match up with this data (time in seconds): ($Target : M = 189.729$, $SD = 100.887$, $Ripple : M = 194.6$, $SD = 105.929$, $Path : M = 312.878$, $SD = 206.606$, $Arrow : M = 152.825$, $SD = 82.377$). The completion time for Arrow is fastest on average while Path is slowest on average (taking more than twice as long). This seems to indicate that the Arrow is easier to get accustomed to and use to effectively search an environment, and Path is slower to understand and to use moment-to-moment.

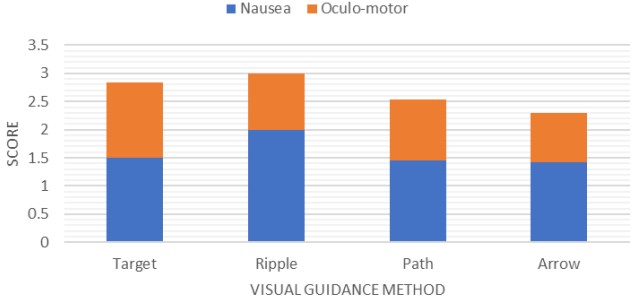

Figure 15: Average scenario completion times (in seconds) with 95% confidence error bars (lower is better). There is a significant difference between Arrow (fastest average), and Path (slowest average).

### 4.2.2 Preferences

We found no statistically significant difference in Preference ($F(3, 24) = 3.65$, $p = 0.302$). Here are the average Preferences (as seen in Fig. 16) (lower is better): ($Target : M = 2.29$, $SD = 0.999$, $Ripple : M = 2.88$, $SD = 1.076$, $Path : M = 2.58$, $SD = 1.1389$, $Arrow : M = 2.25$, $SD = 1.225$). Arrow was on average preferred but was very close in preference to Target, possibly indicating a split between preference of a more direction/angle-based guidance in Arrow and an absolute location guidance in Target.

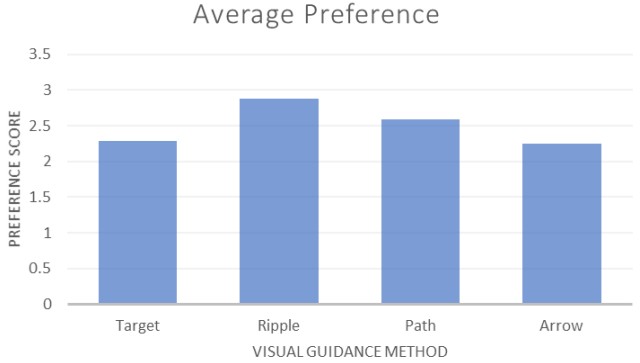

Figure 16: Mean Preference (lower is better). Arrow is lower than Target, but by only a small margin.

### 4.2.3 SSQ

We found no statistically significant difference in SSQ total score ($F(3, 24) = 2.404$, $p = 0.493$), Nausea sub score ($F(3, 24) = 1.451$, $p = 0.694$), or Oculo-motor sub score ($F(3, 24) = 4.274$, $p = 0.233$). Table 2 lists the mean and standard deviations for the SSQ scores, and Fig. 17 shows a bar graph visualization. For simulator sickness, Arrow had the least average effect in general and for all subcategories.

### 4.2.4 UES SF

We found no statistically significant difference in UES SF total score ($F(3, 24) = 3.967$, $p = 0.265$) and all sub scores: FA ($F(3, 24) = 6.745$, $p = 0.08$), AE ($F(3, 24) = 4.432$, $p = 0.218$), PU ($F(3, 24) = 2.86$, $p = 0.414$), and RW ($F(3, 24) = 2.814$, $p = $

Table 2: Mean and standard deviation of SSQ scores and subscores. T=total score, N=Nausea, O=Oculo-motor (lower is better).

| Score | Target | Ripple | Path | Arrow |
|---|---|---|---|---|
| N | M = 2.6 | M = 2.65 | M = 2.4 | M = 2.35 |
| | SD = 1.794 | SD = 2.621 | SD = 2.206 | SD = 2.263 |
| O | M = 2.79 | M = 2.48 | M = 2.46 | M = 2.27 |
| | SD = 1.81 | SD = 1.351 | SD =1.501 | SD = 1.624 |
| Total | M = 2.73 | M = 2.58 | M = 2.42 | M = 2.27 |
| | SD = 3.239 | SD = 3.707 | SD = 3.257 | SD = 3.77 |

Average SSQ Scores

Figure 17: Average scores for SSQ, including sub scores (lower is better). Arrow has lowest scores in total and sub scores.

0.421). Table 3 lists the mean and standard deviations for the UES SF scores, and Fig. 18 shows a bar graph visualization.

Table 3: Mean and standard deviation of UES SF total scores and sub scores (higher is better).

| Score | Target | Ripple | Path | Arrow |
|---|---|---|---|---|
| FA | M = 2.23 | M = 2.21 | M = 2.83 | M = 2.73 |
| | SD = 1.054 | SD =0.861 | SD = 0.735 | SD = 0.927 |
| AE | M = 2.35 | M = 2.31 | M = 2.46 | M = 2.88 |
| | SD = 0.948 | SD = 1.023 | SD = 0.832 | SD = 0.784 |
| PU | M = 2.46 | M = 2.23 | M = 2.56 | M = 2.75 |
| | SD = 0.624 | SD = 0.917 | SD = 0.778 | SD = 0.581 |
| RW | M = 2.5 | M = 2.31 | M = 2.4 | M = 2.79 |
| | SD = 1.054 | SD =0.809 | SD = 0.671 | SD = 0.72 |
| Total | M = 2.42 | M = 2.19 | M = 2.52 | M = 2.88 |
| | SD = 0.795 | SD =0.711 | SD = 0.644 | SD = 0.607 |

Arrow had the highest average effect on overall user engagement, AE, PU, and RW. The only score Arrow was not the highest in was FA, with Path being the highest average effect. This possibly indicates a difference in conditions that allowed or required more attention in Path compared to the others.

### 4.3 Discussion

We developed RealNodes to investigate creating visual guidance methods effective for 360 VR, and to perform a study to determine what effect choice of visual guidance had on preference, simulator sickness, user engagement, and task completion. We successfully implemented RealNodes and four visual guidance methods: Target, Ripple, Path, and Arrow. We completed a comparative evaluation, providing a statistically significant finding that Arrow is faster for users to learn and use, contributing to faster task completion times.

Arrow is unique compared to the other UIs in that it stays active on screen. Participants liked how it smoothly curved towards the nearest waypoint, giving continuous feedback on waypoint locations. Additionally, participants liked how it changed color when they could do WIP. One participant described it as "feeling good for exploration". Another described it as "a combination" of showing

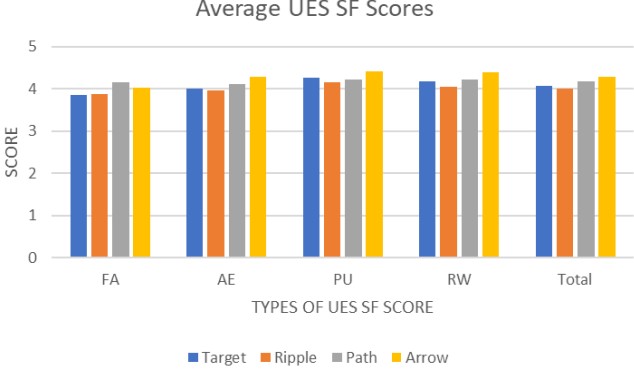

Figure 18: Average scores for UES SF sub scores and total scores (higher is better). Arrow had the highest scores in all categories except FA, where Path has the highest.

direction and location compared to other UIs. Meanwhile, Path enables and disables rendering based on if the user's facing direction is near a waypoint. Participants claimed Path was unclear for how they could navigate and gave less information compared to Arrow.

Trends show Target and Arrow close in Preference, but notably ahead of the others. Some participants said they liked Target over Ripple because it showed exact location of the waypoint and it was less distracting. Participants preferred Arrow over Path because of how it pointed in the exact direction of a waypoint and it was less vague. Since SSQ data was not significantly different and all averages are low relative to the SSQ scale (3 or below), we conclude our UIs had no significant effect on simulator sickness in our scenario. An interesting trend was noticed for UES SF, where averages for all but one sub score had the same ranking, with Arrow performing best. However, FA had Path beat Arrow. FA is described by the authors of the test as indicating "cognitive absorption and flow" and losing track of time [17]. This could be interpreted negatively, meaning the difficulty of using Path required more focus.

## 5 FUTURE WORK

The implications of the results provide new work to be done. Future studies are needed to make generalizable claims about the effect of our navigation and visual guidance cues on user experience. Larger populations and deeper analysis of demographics differences such as gender are needed. Different combinations of visual guidance with navigation technique (point-and-click ray cast, WIP, or automated scenario with no navigation) should be compared to determine if effects are due to guidance method and/or navigation method. Based on our initial findings, we can improve our navigation techniques based on refined hypotheses, such as developing hybrid UIs. Additionally, we can deemphasize current measures and focus on cognitive load and accuracy. These changes in future studies may glean more generalizable, meaningful results for the field of 360 VR.

Further research into authoring tools is needed, especially asset management of videos and animations, making it easier to create scenarios. Some participants indicated a desire to see different environments besides the House Scenario. A variety of scenarios is worth investigating for richer experiences, as well as mitigating learning effects over time, aiding in generalization of future findings.

Our current system for indicating WIP start/end can be improved. Existing visual cues (UI disappears/reappears at the start/end), this was not enough for some participants to quickly and clearly know they were "done" with WIP. Solutions include a progress indicator during the walk or an absolute waypoint getting closer.

There are opportunities for 3D computer vision to be leveraged.

3D reconstruction with a monoscopic 360-video has been explored. Im et al. [9] proposed a promising method using Small Structure from Motion with a consumer 360 camera to generate a dense depth map. It uses a "sphere sweeping" dense matching that looks at overlapping regions on the camera lenses. The video only required small jittery movements for reconstruction to work. Synthetic and real-world benchmarks resulted in low reprojection error compared to bundle adjustment and ground truth. There is also a possibility for novel view synthesis. For RealNodes, we manually created transition videos, which is tedious and discourages complex scenarios. Fusiello and Irsara [8] describe a pipeline that turns two or more uncalibrated images into a "video" sequence of images. In theory this could work on 360-videos to produce novel video transitions in a possible authoring tool. However, epipolar geometry calculation must be handled differently compared to a standard perspective image. An appropriate method needs further investigation.

Some participants were interested in a "full game" like the House Scenario with a larger narrative. They reacted favorably to the background character in the videos, asking "Who is he?" and saying he reminded them of horror/thriller games. RealNodes is designed to present videos based on event triggers, making future scenarios with branching storylines and decision making possible. However, methods of immersive interaction with characters in a 360 VR environment needs further investigation. We look towards considerations for proxemics when filming 360-video, such as considering personal space while actors interact with a 360-video camera [19].

## 6 CONCLUSION

We developed a 360 VR system geared towards navigation and interaction, and to determine if the choice of visual guidance UI in 360 VR exhibited a significant difference in the user's experience. We successfully developed RealNodes to facilitate 360 VR scenarios with explorable and interactive environments. We successfully contribute to the literature a set of four visual guidance UI elements for use in 360 VR to be refined and iterated on in future applications. We additionally provide the results of a comparative evaluation. We were able to successfully determine that using our Arrow visual guidance UI provided a significant benefit to completion time compared to our other designs. We additionally provide the results of a comparative evaluation of our four visual guidance elements, demonstrating that our designs had no significant effect on preference, simulator sickness, or user engagement in the scenario we developed. Emerging 360 VR application types directly benefit from exploring these kinds of UIs. Virtual tours and occupational training software can have a wide variety of locations and paths with less concern of confusing the user. Exercise software can be made using either WIP or another locomotive technique in 360 VR allowing for branching paths that can be telegraphed as the participant moves. New novel kinds of 360 VR games can be designed, delivering an immersive environment that is easy to navigate compared to traditional methods. Our exploration of navigation in 360 VR experiences provides feedback for future research, reducing the need to design visual guidance UI from scratch, allowing more time for the research to focus on the remaining challenges of developing novel, immersive, and interactive 360 VR worlds.

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
