# OpenReview forum: "RealNodes: Interactive and Explorable 360$^{\circ}$ VR System with Visual Guidance User Interfaces"
_graphicsinterface.org/Graphics_Interface/2020/Conference — Submitted to GI 2020_

### Official Review · AnonReviewer2 · 2020-04-19
**Well executed, but unclear contribution**

**Rating:** 5
**Confidence:** 3

**Review:**

#== SUMMARY ==#

The authors present RealNodes. VR users can freely look around in a pre-recorded 360 video. Multiple 360 videos are shot at different key locations. Users can navigate between them using gaze direction and walking in place to simulate moving to different locations in space. For each 360 video, particular gaze directions are mapped to different pre-defined paths available for navigation. The system provides visual guidance to communicate, which directions the users can move to from their position. The authors compared four different visual aids and found significant differences between the techniques in terms of completion time and preference. In the paper, the authors describe the implementation of the system and report the study results.

#== REVIEW ==#

The authors present a well made system with a nicely executed study. The paper is well written and easy to follow. However, there are issues with the novelty and generalizability of the results. In the following, I would like to elaborate on those issues.

The amount of related work is somewhat sparse, but there are some places in which related work is incorporated well to inform the design. For instance, the guidance techniques are inspired by previous work. However, other parts, like the claimed "novel additions" or "novel changes" in section 3 are not backed by related work. In fact, those claimed "novelties" are just specific ways of implementing the desired system by customizing existing assets. Therefore, they have no novelty from a research perspective and, again, not positioning them in literature makes those claims entirely unfounded. For instance, using three textures as the environment map and combining them, e.g., through blending is described as novel, but this is just a specific way of implementing the desired effect. Therefore, while the implementation is well made, there is no contribution from a technical perspective. The descriptions about the implementation details can be shortened and the concept can be explained in more general terms than describing it around Unity3D assets. Furthermore, implementation details like for instance preloading videos into memory are not essential for reproducibility.
That said, the description about synchronizing the walking speed (i.e., the oscillation of the HMD) with the video playback is the most interesting of the presented ideas. However, this part is not fleshed out and ends up as another minor addition or implementation detail, even though it might have some potential to be explored more in depth.

The study is well designed and executed. However, it is hard to generalize or draw conclusions from the results. While there are significant differences, those are bound to the specific design of the four techniques and are prone to many confounding variables. For instance, "path" almost looks like an incomplete implementation of a visual aid compared to arrow. The significant factor might be the fact that the arrow is more integrated and curved while the path is always straight along the viewing direction, which might be misleading. The arrow (i.e., including the arrow tip) in itself might have a very insignificant role, i.e., having a curved path might have yielded the same results. One simple way of mitigating this would be to rename the conditions, e.g., curved versus non-curved. However, the issue remains that there are many confounding variables and that the results are hard to generalize from.
A closely related issue is that the techniques provide different amounts of information, making them hard to compare. This is again most apparent in the arrow versus path case. The arrow has all information to guide the user precisely to the closest waypoint, whereas the path only provides binary information, i.e., whether there is a waypoint currently viewed. With the increasing number of waypoints, it might even never be inactive, basically showing no information. As of now, the arrow implementation looks like a heavily improved path visualization as it contains a curved path plus an arrow tip. One approach to have those two techniques comparable would be to compare a curved path with a rotated arrow so as to make them display the same amount of information. Alternatively, the path could be aligned with the footage (e.g., like in Google Street View) instead of just being overlayed as an image. As of now, I believe that the path condition invalidates most of the results as we cannot derive that paths are in fact inferior to arrows as claimed in the paper. Lastly, the ripple visualization is very specific and distracting (as also pointed out by participants). Instead of a subtle ripple effect, it is very opaque and occludes the actual region of interest. While the other techniques can potentially be made comparable, the ripple technique has a lot of parameters (opacity, size, shape, amount of distortion etc) making it hard to compare with the other techniques, which are more basic. In more general terms, the goal of the study has to be made more clear. What are preferences based on? Is it a trade-off of the amount of information, distraction and aesthetics?

In summary, the system is well implemented and the study is well executed. However, it is hard to tell what the contribution of this paper is, because neither is there technical novelty, nor do the results have clear implications. Therefore, I slightly lean towards rejection of this work.

---

### Official Review · AnonReviewer1 · 2020-04-20
**Unclear study on visual guidance UIs in 360 degree VR environment**

**Rating:** 5
**Confidence:** 3

**Review:**

This paper explores the design of user interfaces (UIs) for visual guidance in the context of interactive and navigate-able 360 degree virtual reality (VR) systems. They present and describe a software system that enables the interactive and navigate-able 360 degree VR environment, within which they implemented four visual guidance UI techniques.

Overall the research objective seems interesting, but I would have appreciated a clearer overall presentation of the research and the specific contributions, and a clearer motivation of the research gap. I also believe that the the study design holds some potential flaws that need addressing.

The introduction mixes in some lists of related work that would be better suited in the respectively named section; instead, a clear and focused motivation of the research gap being explored here would benefit the paper. The authors motivate the system a little bit, but not so much the visual guidance UI.

I think it would be very helpful as well if the introduction provided a summary of the paper's contributions. This would also help to assess whether the number of pages is well matched to the size of the contribution.
Although the end of the introduction seems to focus more on the study (as does the Discussion), I am pretty sure that the authors are presenting the RealNodes software system as part of the main contributions, as the paper goes into a lot of detail on this - to clarify, I agree that this can represent a contribution, however the section in which this is described could use some more general / abstract overview information before deep-diving into the specifics, and some more "sign-posting" in the text's structure. In particular, I think that "3.4 - Visual Guidance User Interface Methods" should get more focus compared to the other aspects, as they represent the focus of the study presented in the paper, as opposed to the interaction objects and scenarios that are then described in detail in 3.5 & 3.6.
A clearer summary of the systems' overall benefits as well as a bit more motivation on why *these* four UI visual guidance techniques were chosen for the system would also be helpful.

The related work addresses navigation techniques, wayfinding guidance, visual transitions, and interactive elements. These are largely presented as as a list; the paper would be stronger if it provided a bit more of a summary that outlines the research gap addressed by the authors.

Study:
In an experimental within-subjects user study (N=24), the paper explores effects of four different UI techniques on engagement, simulator sickness, and completion time. Some general notes:
- participants' ages are not reported. gender distribution could be a bit better. prior VR experience not reported.
- UI techniques were counterbalanced, but were the hiding spots counter-balanced as well?
- effect sizes are not reported
- some of the descriptive data reported in-text would be better suited for display in a table
- what is the scale of the SSQ, does it only go up to 3.5 as indicated by Fig.17? If so, then the nausea scores aren't that great for any of the conditions.
- similarly, what is the scale range of the UES SF?

Finally, and most importantly, the discussion then states: "Arrow is unique compared to the other UIs in that it stays active on screen."
Before this statement, it was not clear to me that the Path technique was not visible continuously.
If only Arrow is continuously shown, then the study design seems unbalanced, comparing 1 continuous wayfinding techniques with 3 discrete target marking techniques. That Arrow then leads to significantly faster completion times hardly seems surprising.

Overall, I find that the potentially unbalanced design and the missing information in the study (particularly the effect sizes) make it very hard to judge what is being investigated here. Without addressing these issues, I do not think that this submission is quite ready for publication.

Minor points:
- "reignited" -> "re-ignited"
- "[21] [2]" -> \cite{bibentry21, bibentry2} -> [2, 21] (can use a parameter for the package to tell LaTeX to auto-sort the citations)
- "demonstrates how we build upon and differentiates from prior research" -> "and differentiate our work from..."
- Fig. 1 is a nice overview but also to a degree redundant with Fig.5, Fig.6, Fig. 7, and Fig.8

---

### Official Review · AnonReviewer3 · 2020-04-23
**Great idea and implementation**

**Rating:** 7
**Confidence:** 4

**Review:**

This paper proposes a VR video system which uses "nodes" to enable navigation in a video-based 3D environment. The user sees real 360-degree imagery captured at each node and can navigate between them. The authors present their implementation and a user study on different navigation/cueing techniques.

Foremost, I think this idea is immensely cool and I appreciate the authors for realizing it into a fully functional system. Like old adventure games (e.g. Myst), this approach enables users to navigate a virtual environment consisting of pre-rendered content (in this case: real videos which would be hard or impossible to replicate as pure renderings). The system comprises a lot of neat ideas - walking-in-place locomotion (WIP), composited 3D videos (green-screen), interactivity with video cut-outs, and navigation between nodes. It really feels like the design of an old adventure game updated for VR.

On the negative side, I found the study somewhat lacking. It lacks external validity as the authors did not compare against any existing baseline approaches (e.g. direct teleportation between nodes)/ Since teleportation is one of the most common VR locomotion techniques, omitting it for comparison seems like a significant limitation. It's a bit odd to state that Arrow was "more than two times faster" in the introduction; it rather seems like the Path condition is the outlier which performed significantly worse than the other conditions. The study also misses a good opportunity to gather subjective feedback from users on the nodes and the navigation scheme in general, i.e. whether it was intuitive or easy for users to understand and use.

The paper should also cite some existing work on nodal VR experiences, for example:

- M. P. Jacob Habgood, D. Moore, D. Wilson and S. Alapont, "Rapid, Continuous Movement Between Nodes as an Accessible Virtual Reality Locomotion Technique," 2018 IEEE Conference on Virtual Reality and 3D User Interfaces (VR), Reutlingen, 2018, pp. 371-378.

Despite the study, I think the system itself is useful and novel, and I would argue for acceptance provided the authors tone down their claims about the study and its results.

---

### Meta-Review · Area_Chair1 · 2020-04-23

**Recommendation:** Reject
**Confidence:** 3

**Metareview:**

This submission is very much a borderline one, making it very hard for me to make a clear recommendation.
R1 and R2 tended towards rejection due to limitations with the study design and lack of clarity in terms of contribution. In contrast, R3 argues for acceptance due to the novel and exciting overall idea (but also criticises the study design).

Overall, being forced to make a choice, I am recommending rejection, however I encourage the authors to iterate on and continue this work: with a clarified contribution and a more polished study, I do think that this work would benefit this (and related) research communities.

---

### Decision · Program_Chairs · 2020-04-25

Reject